# Problematic Smartphone Use and Its Associations with Sexual Minority Stressors, Gender Nonconformity, and Mental Health Problems among Young Adult Lesbian, Gay, and Bisexual Individuals in Taiwan

**DOI:** 10.3390/ijerph19095780

**Published:** 2022-05-09

**Authors:** Mei-Feng Huang, Yu-Ping Chang, Wei-Hsin Lu, Cheng-Fang Yen

**Affiliations:** 1Department of Psychiatry, School of Medicine, College of Medicine, Kaohsiung Medical University, Kaohsiung 80708, Taiwan; 1010236@kmuh.org.tw; 2Department of Psychiatry, Kaohsiung Medical University Hospital, Kaohsiung 80756, Taiwan; 3School of Nursing, The State University of New York, University at Buffalo, Buffalo, NY 14260, USA; yc73@buffalo.edu; 4Department of Psychiatry, Ditmanson Medical Foundation Chia-Yi Christian Hospital, Chiayi 60002, Taiwan; 5College of Professional Studies, National Pingtung University of Science and Technology, Pingtung 91201, Taiwan

**Keywords:** smartphone, psychological well-being, sexual minority, stigma, minority stress

## Abstract

Smartphones are a necessity for many people; however, problematic smartphone use (PSU) may negatively influence people’s mental health. Using multivariate linear regression analysis, the study examined the associations of sexual minority stressors [namely perceived sexual stigma from family members, sexual orientation microaggressions (SOMs), and internalized sexual stigma] and gender nonconformity with PSU severity as well as the associations of PSU with depression and anxiety in young adult lesbian, gay, and bisexual (LGB) individuals. This cross-sectional survey study recruited 1000 young adult LGB individuals (500 men and 500 women). PSU severity was assessed using the Smartphone Addiction Inventory. The experiences of perceived sexual stigma from family members, SOMs, and internalized sexual stigma and the levels of gender nonconformity, depression, and anxiety were assessed. The results indicated that perceived sexual stigma from family members, SOMs, internalized sexual stigma in the dimensions of social discomfort and identity and gender nonconformity were significantly associated with PSU severity in LGB individuals. Moreover, PSU was significantly associated with depression and anxiety in LGB individuals. The findings highlight the significance of developing strategies for the prevention and early detection of PSU and sexual minority stress in LGB individuals.

## 1. Introduction

### 1.1. Problematic Smartphone Use and Its Associations with Health Problems

Griffiths firstly proposed the concept of technological addictions and operationally defined it as a behavioral addiction that involves human-machine interaction and is non-chemical in nature [1]. The Diagnostic and Statistical Manual of Mental Disorders, 5th edition (DSM-5) proposed the diagnostic criteria of Internet gaming disorder to define the addiction to Internet gaming under the conditions for further study of Section III [2]. Smartphone addiction or problematic smartphone use (PSU) is another technological addiction that warranted further examination [3,4,5,6,7]. Smartphones are an epoch-making invention. Smartphones can assist people in their daily lives by speeding up online information transfer, services, and interpersonal interaction as well as providing entertainment, such as playing games, watching videos, and taking pictures. However, studies have found that PSU is consistently related to mental health problems (e.g., depression, anxiety, chronic stress, and low self-esteem) [8], physical discomfort (e.g., back, neck, and wrist pain) [9], and distracted driving in young adults [10]. Research has suggested that smartphone addiction or PSU should be conceptualized as a multi-dimensional construct. For example, Lin et al. proposed that smartphone addiction contains the dimensions of compulsive behavior, functional impairment, withdrawal and tolerance [3]. Kwon et al. proposed that smartphone addiction contains the dimensions of daily-life disturbance, positive anticipation, withdrawal, cyberspace-oriented relationship, overuse, and tolerance [11]. Both Lin et al. [3] and Kwon et al. [11] identified tolerance, withdrawal and function impairments as the essential characteristics of PSU, indicating that PSU shares the core symptoms with substance use disorders and Internet gaming disorder in DSM-5 [2] and called for further study on PSU in people of various groups.

### 1.2. PSU in Lesbian, Gay, and Bisexual Individuals

Smartphone use plays an important role in the daily lives of young adult lesbian, gay, and bisexual (LGB) individuals for several reasons. First, the public has stigmatizing attitudes toward these individuals [12]; in such unfriendly social environments, many LGB individuals conceal their sexual orientation and search for online social support. LGB individuals might rely on social support obtained on the internet and make more online friends than heterosexual individuals [13,14]. Second, as LGB individuals constitute a small group, chance encounters with partners are less likely in the daily social activities of these individuals compared with heterosexual individuals [15]; therefore, online dating and seeking sexual partners are common among LGB individuals [16,17]. However, the use of smartphones and the internet has a reinforcing effect, increasing their use [8] and thereby increasing the possibility of PSU among LBG individuals. A study found that problematic social media use among LGB young adults is associated with low social support and depression symptoms [18]. Examining PSU-related factors and the associations of PSU with mental health problems in young LGB adults may provide a reference for developing intervention strategies to reduce the PSU risk.

### 1.3. PSU and Sexual Minority Stressors

Social and psychological stressors are significantly associated with PSU severity [19,20]. According to minority stress theory [4], LGB individuals may experience several types of unique stressors as they are living in a social environment full of prejudice and stigma rooted in heterosexism. In addition to homophobic bullying victimization [21], common sexual minority stressors include perceived sexual stigma from close people, such as family members, sexual orientation microaggressions (SOMs), and internalized sexual stigma. Familial sexual stigma indicates the ignorance, prejudice, and discrimination of family members toward LGB individuals [22,23,24], which can result in negative health outcomes among LGB individuals [22,23]. SOMs indicate commonplace and brief daily behavioral, verbal, and environmental contempt reflecting hostile or derogatory slights and insults to LGB individuals for their sexual orientation [25,26]. SOMs increase the risk of mental health problems in LGB individuals [27,28]. Internalized sexual stigma means that LGB individuals perceive and internalize societally stigmatizing attitudes toward their sexual orientation as part of their self-image [29]. Internalization of sexual stigma may compromise relationship well-being [30] and their intention to adopt safe sex among LGB individuals [31,32]. Homophobic bullying victimization in childhood can predict PSU severity in young adulthood among gay and bisexual men [33], and problematic social media use is associated with internalized stigma among young LGB adults [18]. No study has examined the associations of perceived sexual stigma from family members, SOMs, and internalized sexual stigma with PSU severity simultaneously in LGB individuals.

### 1.4. PSU and Gender Nonconformity

In addition to the aforementioned sexual minority stressors, the association of gender nonconformity with PSU severity in LGB individuals warrants study. Gender nonconformity is defined as an individual’s behavioral, cultural, and psychological traits that do not match with the socially expected traits of the gender associated with the individual’s biological sex [34,35]. A literature review reported that gender nonconformity is significantly related to negative social experiences in both heterosexual and LGB individuals, such as sexuality-related bullying victimization [36]. It is supposed that gender nonconformity increases people’s conjecture about individuals’ sexual orientation and repulsion due to the stereotype of the gender role. Whether gender nonconformity is significantly associated with PSU severity in LGB individuals warrants examination.

### 1.5. Aims of This Study

This cross-sectional survey study had two aims. First, we examined the associations of various sexual minority stressors (e.g., perceived sexual stigma from family members, SOMs, and internalized sexual stigma) and gender nonconformity with PSU severity in LGB individuals by controlling for the effects of gender, age, and sexual orientation. Second, we examined the associations of PSU with depression and anxiety by controlling for other factors. Although a recent study on 29,712 individuals in South Korea showed that smartphone users’ age and gender do not contribute considerably to predicting the PSU level [37], other studies have found that women have more severe PSU than men [38,39]. Moreover, research has revealed that PSU severity differs across various sexual orientation groups [40]. We hypothesized that after controlling for other factors, all three sexual minority stressors and gender nonconformity were significantly associated with PSU severity and that PSU severity was significantly associated with depression and anxiety in LGB individuals.

## 2. Materials and Methods

### 2.1. Participants and Procedure

This cross-sectional survey enrolled participants by posting advertisements on social media sites frequently used by young adults in Taiwan, including Facebook (Meta Platforms, Inc., Menlo Park, CA, USA), Twitter (Twitter, San Francisco, CA, USA), and LINE (LINE Corporation, Tokyo, Japan; a direct messaging app); the Bulletin Board System (InterSoft International, Inc., Katy, TX, USA; a popular application dedicated to sharing or exchange of messages on a network); and the homepages of three health promotion and counseling centers for LGB individuals from August 2018 to July 2020. We also encouraged people to share the advertisements with their LGB friends and invite them to participate in this study. The inclusion criteria were (i) age between 20 and 30 years, (ii) being lesbian, gay, or bisexual, and (iii) residing in Taiwan. Adults who had impaired intellect, used addictive substances and alcohol, or had severe mental illnesses, such as schizophrenia and cognitive disorders due to brain injury and had difficulties in understanding the study’s purpose or in completing the study questionnaire were excluded from this study. Those who intended to participate in this study could call the research assistants. The assistants reconfirmed eligibility on the phone and scheduled an assessment time. Informed consent was obtained from all participants prior to assessment. Participants completed the study questionnaire individually in the study room of a university affiliated hospital. A total of 1000 LGB participants (500 women and 500 men) were enrolled in this study. This study was approved by the Institutional Review Board of Kaohsiung Medical University Hospital (KMUHIRB-F(II)-20180018).

### 2.2. Measures

#### 2.2.1. PSU

We used the 26-item Smartphone Addiction Inventory (SPAI) [3] to assess participants’ PSU severity, including compulsive use, functional impairment, withdrawal, and tolerance, in the month preceding the study period. Each item was rated on a 4-point Likert-type scale, with the score ranging from 1 (*strongly disagree*) to 4 (*strongly agree*). A higher total score indicates a greater tendency of PSU. The SPAI has acceptable psychometric properties [3].

#### 2.2.2. Demographic and Sexual Orientation Factors

We collected information on participants’ gender, age, sexual orientation (homosexual or bisexual), and gender nonconformity. Participants self-rated their level of gender role self-identity on a 9-point Likert-type scale, with the score ranging from 1 (*extreme femininity*) to 9 (*extreme masculinity*) [41]. Responses of gay and bisexual men were reversely scored. A higher score indicates a higher gender nonconformity level.

#### 2.2.3. Perceived Sexual Stigma from Family Members

We used the 10-item homosexuality subscale of the Chinese version of the HIV and Homosexuality Related Stigma Scale (HHRS), which has acceptable psychometric properties [42], to assess perceived stigmatizing attitudes toward homosexuality from family members (e.g., “My family believes that gay/lesbian individuals are promiscuous” and “My family unwillingly accepts gay/lesbian individuals”). Each item was rated on a 4-point Likert-type scale, with the score ranging from 1 (*strongly disagree*) to 4 (*strongly agree*). A higher total score indicates a higher level of perceived sexual stigma from family members [42].

#### 2.2.4. Sexual Orientation Microaggression

We used the 19-item traditional Chinese version [28] of the Sexual Orientation Microaggression Inventory (SOMI) [43] to assess participants’ experiences of SOMs over the last 6 months, including anti-LGB attitudes and expressions (e.g., “You heard someone talk about ‘the gay lifestyle’”), denial of homosexuality (e.g., “A family member expressed disappointment about you being gay, lesbian, or bisexual”), heterosexualism (e.g., “You were told that you were overreacting when you talked about a negative experience you had because of your sexual orientation”), and societal disapproval (e.g., “Someone said homosexuality is a sin or immoral”). Each item was rated on a 5-point Likert-type scale, with the score ranging from 1 (*not at all*) to 5 (*almost every day*). A higher total SOMI score indicates higher exposure to SOMs. Both the original [43] and traditional Chinese versions [44] of SOMI have acceptable psychometric properties. The Cronbach’s α value in the present study was 0.90.

#### 2.2.5. Internalized Sexual Stigma

We used the 17-item Measure of Internalized Sexual Stigma for Lesbians and Gay Men (MISS-LG) to assess participants’ level of internalized sexual stigma on 3 subscales, namely social discomfort (e.g., “At university/at work, I pretend to be heterosexual [pretending to be attracted to women/men or showing typically male/female interests]”), sexuality (e.g., “Gay/lesbians can only have flings/one-night stands”), and identity (e.g., “Sometimes I think that if I were heterosexual, I could be happier”) [45]. Each item was rated on a 5-point Likert scale, with the score ranging from 1 (*strongly disagree*) to 5 (*strongly agree*). A higher total subscale score indicates a higher level of internalized sexual stigma. Both the original [45] and traditional Chinese versions [46] of MISS-LG have acceptable psychometric properties.

#### 2.2.6. Depression

Participants’ severity of depression in the previous 1 month was measured using the Taiwanese version [47] of the Center for Epidemiological Studies—Depression Scale [48] (CES-D). Participants self-rated their frequencies of 20 depressive symptoms on a 4-point scale. A higher total score indicates more severe depression.

#### 2.2.7. Anxiety

Participants’ severity of current anxiety was measured using the state subscale [49] of the State-Trait Anxiety Inventory [50] (STAI-S). Participants self-rated their severities of 20 anxiety symptoms on a 4-point scale. A higher total score indicates more severe anxiety.

### 2.3. Statistical Analysis

We analyzed the data using IBM SPSS software (version 20.0; IBM, Armonk, NY, USA). Participants’ demographics, sexual orientation factors, and various sexual minority stressors were analyzed and are presented as mean (standard deviation [SD]) and frequency (percentage). The absolute values of skewness and kurtosis of continuous variables were <2 and were normally distributed [51]. Pearson correlations were carried out to calculate the bivariate correlations between the studied variables. To avoid the problem of collinearity, we used 4 multivariate linear regression models to examine PSU-related factors (Figure 1). Model I examined the associations of gender nonconformity (independent variable) with PSU (dependent variable). Model II examined the association of perceived sexual stigma from family members with PSU. Model III examined the association of SOMs with PSU. Model IV examined the associations of three dimensions of internalized sexual stigma with PSU. All four models controlled for the effects of gender, age, and sexual orientation. The associations of PSU with depression and anxiety were further examined through stepwise multivariate linear regression analysis, which controlled for the effects of sexual minority stressors, gender nonconformity, sexual orientation, and demographics. Stepwise multivariate linear regression analysis could reduce the collinearity level. A *p*-value of 0.05 was used to indicate significance in all statistical tests.

## 3. Results

Table 1 summarizes the data on PSU, demographics, sexual orientation factors, and various sexual minority stressors of the 1000 participants [mean (SD) age: 24.6 (3.0) years]. Among the participants, 29.5% were bisexual women, 20.5% were lesbians, 13.5% were bisexual men, and 36.5% were gays; the mean (SD) level of gender nonconformity was 4.5 (1.5) years. The mean scores (SD) for PSU, perceived sexual stigma from family members, and SOMs were 61.8 (14.5), 26.6 (6.5), and 42.0 (11.6), respectively. The mean scores (SD) for the social discomfort, sexuality, and identity dimensions of the MISS-LG were 16.6 (6.0), 8.9 (3.3), and 9.9 (4.2), respectively.

Table 2 provides the bivariate correlations between the studied variables. Problematic smartphone use was significantly positively correlated with the following variables: perceived stigma from family members (r = 0.219; *p* < 0.001), sexual orientation microaggression (r = 0.202; *p* < 0.001), and internalized sexual stigma dimensions of social discomfort (r = 0.195; *p* < 0.001), sexuality (r = 0.139; *p* < 0.001) and identity (r = 0.215; *p* < 0.001).

Table 3 presents the results of the four multivariate linear regression models examining the associations of various sexual minority stressors and gender nonconformity with PSU. Condition indices of Model I, Model II and Model II were 24.609, 27.731 and 27.745, indicating no problem of collinearity (<30). However, the condition index of Model IV was 31.054, indicating that the problem of collinearity needed to be noticed. The results of Model I demonstrated that after controlling for gender, age, and sexual orientation, gender nonconformity was positively associated with PSU. The results of Model II and Model III demonstrated that after controlling for the effects of demographics, sexual orientation, and gender nonconformity, perceived sexual stigma and SOMs were positively associated with PSU. The results of Model IV demonstrated that the internalized sexual stigma dimensions of social discomfort and identity, but not sexuality, were positively associated with PSU.

Table 4 presents the results of stepwise multivariate linear regression analysis of the associations of PSU with depression and anxiety. The results indicated that after controlling for the effects of sexual minority stressors, gender nonconformity, and other factors, PSU was significantly associated with depression and anxiety in LGB individuals.

## 4. Discussion

The present study found that after controlling for the effects of gender, age, and sexual orientation, perceived sexual stigma from family members, SOMs, internalized sexual stigma, and gender nonconformity was significantly associated with PSU severity in LGB individuals; moreover, PSU was significantly associated with depression and anxiety in LGB individuals. This is the first study to examine the associations of various sexual minority stressors and gender conformity with PSU in LGB individuals. Smartphones have a crucial role in LGB individuals’ social lives [13,14,15,16,17]; therefore, the present study results can be used to develop risk-group specific interventions to prevent PSU in LGB individuals.

### 4.1. PSU, Perceived Sexual Stigma from Family Members, SOMs and Internalized Sexual Stigma

Perceived sexual stigma from family members and SOMs may increase PSU severity in LGB individuals through several mechanisms. Sexual stigma from family members may manifest through familial negative attitudes and various negative behaviors of family members, including disdaining the sexual orientation of homosexuality or bisexuality and sexual orientation-related rejection and harassment [22,23,24]. LGB individuals may conceal their identity from their families to avoid familial stigmatizing treatment [52]; they may turn to sources outside their families for support regarding their sexual orientation, including online friends made through smartphones. LGB individuals who experience SOMs may feel insulted and unsafe; this feeling may manifest briefly and subtly. Especially, LGB individuals find it difficult to communicate with the enactors about what they feel regarding the insults in SOMs because the enactors often view their own behavior as harmless and unremarkable [53]. LGB individuals may naturally reduce interactions with people in real life to avoid facing SOMs and may increase online social interactions through smartphones. LGB individuals who face SOMs may also avoid seeking treatment for PSU or other psychiatric disorders because of worrying about encountering SOMs from therapists. Delaying treatment for PSU or other psychiatric disorders may further exacerbate PSU.

Because of internalized sexual stigma, LGB individuals may anticipate social rejection, consider sexual stereotypes to be self-relevant, and believe that they are devalued by others [54,55]; difficulties in social relationships may ensue [30]. LGB individuals with internalized sexual stigma may spend more time using smartphones to look for approval on the internet world and assert their self-identity. The internet provides LGB individuals with anonymous and convenient environments to develop relationships with others and feel supported and safe [13,14]. However, LGB individuals may experience sexuality-related cyberbullying [56] and online SOMs [57] prevalent on the internet. LGB individuals with PSU may spend considerable time on the internet; whether they have a higher risk of experiencing sexuality-related stigma in both real and virtual worlds and the ensuing adverse impacts on health warrants further study.

Moreover, perceived sexual stigma from family members [22,23], SOMs [27,28], and internalized sexual stigma [58,59] may all compromise LGB individuals’ mental health. LGB individuals may use smartphones to search for comfort in the internet world. However, the reinforcing effect of internet and smartphone use may aggravate negative self-identity, loneliness, and negative emotions among LGB individuals [60], forming a vicious circle between PSU and mental health problems.

The present study found that the internalized sexual stigma dimensions of social discomfort and identity, but not sexuality, were positively associated with PSU in LGB individuals. The sexuality dimension describes the pessimistic attitude toward intimate gay or lesbian relationships and sexual behaviors [45]; the identity dimension describes the negative self-attitude as an LGB individual [45] and the social discomfort dimension describes the fear of public identification as a lesbian or gay person in social contexts and the fear of disclosure of LGB identity in private and professional lives [45]. LGB individuals may communicate their concerns about sexual orientation identity and social discomfort on the internet, whereas the internalized sexual stigma regarding sexuality may decrease the motivation to seek potential sexual contact using smartphone apps. The associations between various dimensions of internalized sexual stigma and PSU may thus vary in LGB individuals.

### 4.2. PSU and Gender Nonconformity

In the present study, gender nonconformity was significantly associated with PSU severity in LGB individuals. Gender nonconformity has been identified as a risk factor for sexuality-related bullying [21,36], expectations of rejection due to sexual orientation [61], and mental health problems in LGB individuals [62,63,64,65]. LGB individuals with gender nonconformity may spend considerable time using the internet through smartphones for social interaction and entertainment instead of virtually interacting with others to avoid social rejection and discrimination. In this study, gender nonconformity was significantly associated with PSU, which was independent of various sexual minority stressors. The result indicated that health professionals should pay attention to the role of gender nonconformity in PSU among LGB individuals.

### 4.3. PSU, Depression and Anxiety

In this study, after controlling for the effects of sexual minority stressors and gender nonconformity, PSU was significantly associated with depression and anxiety in LGB individuals. Although temporal relationships between PSU and depression and anxiety could not be determined in this study, the findings highlight the value of developing strategies for the prevention and early detection of PSU in LGB individuals. A recent study developed an intervention program that was delivered through a smartphone application, and the intervention program comprised goal setting, personalized feedback, mindfulness, and behavioral suggestion; the intervention resulted in a reduction in self-reported PSU [66]. The light-touch smartphone-delivery of the intervention may be suitable for LGB individuals who are concerned about privacy and structural stigma in medical units or psychological clinics. Because of significant associations between sexual minority stressors and PSU, interventions that enhance the understanding of LGB culture and awareness of prejudices toward LGB individuals in schools, workplaces, families, and society are essential to reduce sexual minority stress for LGB individuals [67,68].

### 4.4. Limitations

This study has several limitations. First, in this study, participant self-reported data were collected; therefore, shared-method variance might have occurred. Second, the cross-sectional study design limited our ability to determine temporal relationships among PSU, sexual minority stress, and mental health problems. Third, the study participants were aged between 20 and 30 years; therefore, whether the study results can be generalized to LGB individuals of other age groups should be examined. Fourth, this study did not inquire whether participants identified their gender as transgender, gender nonbinary, and genderqueer; therefore, the influence of gender minority stress could not be determined. Last, we did not survey the purposes of smartphone use. LGB individuals may use smartphones for various purposes, for example, exchanging messages, searching for partners, playing games, and watching videos. Various purposes of smartphone use may contribute to various levels of reinforcements and severities of PSU.

## 5. Conclusions

Sexual minority stressors (e.g., perceived sexual stigma from family members, SOMs, and internalized sexual stigma) and gender nonconformity were significantly associated with PSU severity in LGB individuals. PSU was significantly associated with depression and anxiety in LGB individuals. Mental health professionals should routinely assess PSU levels and their impacts on LGB individuals’ mental health and social lives, especially those with PSU-related factors. Intervention programs are required to reduce sexual minority stress and PSU among LGB individuals. Given that total acceptance is far from complete due to the continued prevalence of generally unwarranted negative familial and cultural stereotypes, further efforts to modify outdated beliefs about LGB individuals are warranted. Constitutions, laws and anti-discrimination policies at the national level that include protection from discrimination on the grounds of sexual orientation are of fundamental importance [67]. Broadening the understanding of LGB culture and awareness of prejudices toward LGB individuals in educational settings, workplaces, and family environments are also the necessary steps to help reduce sexuality-related stigma [67,68]. Public health strategies addressing attitudes toward sexual orientation and promoting the changes in attitudes toward sexual minorities among the general population may contribute to diverse affirmative cultural scripts regarding LGB individuals’ lives [67,69].

## Figures and Tables

**Figure 1 ijerph-19-05780-f001:**
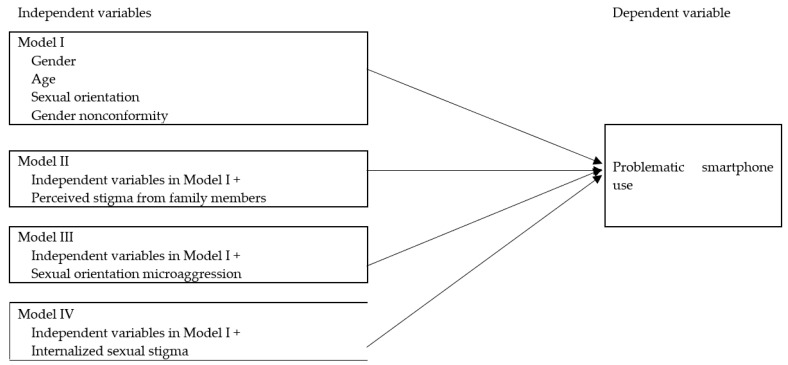
Four Multivariate Linear Regression Models Examining the Factors Related to Problematic Smartphone Use.

**Table 1 ijerph-19-05780-t001:** Participants’ characteristics (N = 1000).

Variables	*n* (%)	Mean (SD)	Range	Cronbach’s α
Gender				
Female	500 (50)			
Male	500 (50)			
Age (years)		24.6 (3.0)	20–30	
Sexual orientation				
Bisexual women	295 (29.5)			
Lesbians	205 (20.5)			
Bisexual men	135 (13.5)			
Gays	365 (36.5)			
Level of gender nonconformity		4.5 (1.5)	1–9	
Problematic smartphone use on the SPAI		61.8 (14.5)	26–101	0.94
Perceived family stigma on the homosexuality subscale of the HHRS		26.6 (6.5)	10–40	0.92
Sexual orientation microaggressions on the SOMI		42.0 (11.6)	19–79	0.90
Internalized sexual stigma on the MISS-LG				
Social discomfort		16.6 (6.0)	7–34	0.86
Sexuality		8.9 (3.3)	5–22	0.67
Identity		9.9 (4.2)	5–23	0.82
Depression on the CES-D		18.8 (11.2)	0–57	0.93
Anxiety on the STAI-S		40.8 (12.7)	20–79	0.89

CES-D = Center for Epidemiological Studies—Depression Scale; HHRS = HIV and Homosexuality Related Stigma Scale; MISS-LG = Measure of Internalized Sexual Stigma for Lesbians and Gay Men; STAI-S = State subscale of the State-Trait Anxiety Inventory; SOMI = Sexual Orientation Microaggression Inventory; SPAI = Smartphone Addiction Inventory.

**Table 2 ijerph-19-05780-t002:** Bivariate correlation matrix between studied variables.

Variables					R					
	1	2	3	4	5	6	7	8	9	10
1. PSU	1									
2. Gender	0.035	1								
3. Age	0.037	0.059	1							
4. Sexula orientation	−0.039	0.323 ***	0.176 ***	1						
5. Gender nonconformity	0.043	−0.241 ***	0.011	0.126 ***	1					
6. Perceived family stigma	0.219 ***	0.081 *	0.063 *	0.001	−0.017	1				
7. SOMs	0.202 ***	0.107 **	0.017	0.014	0.002	0.400 ***	1			
8. ISS: social discomfort	0.195 ***	0.282 ***	0.059	−0.093 **	−0.163 ***	0.278 ***	0.147 ***	1		
9. ISS: sexuality	0.139 ***	0.628 ***	0.017	0.025	−0.189 ***	0.152 ***	0.123 ***	0.599 ***	1	
10. ISS: identity	0.215 ***	0.206 ***	0.005	−0.074 *	−0.009	0.213 ***	0.180 ***	0.630 ***	0.519 ***	1

ISS = Internalized sexual stigma; PSU = Problematic smartphone use; SOMs = Sexual orientation microaggressions. * *p* < 0.05; ** *p* < 0.01; *** *p* < 0.001.

**Table 3 ijerph-19-05780-t003:** Factors related to problematic smartphone use: Multivariate linear regression analysis.

Variables	Problematic Smartphone Use			
	Model I	Model II	Model III	Model IV
	B (SE)	B (SE)	B (SE)	B (SE)
Gender	2.184 (1.013) *	1.604 (0.993)	1.467 (1.000)	−0.534 (1.298)
Age	0.224 (0.155)	0.156 (0.152)	0.209 (0.152)	0.171 (0.153)
Sexual orientation	−2.355 (1.014) *	−2.090 (0.992) *	−2.161 (0.995) *	−0.841 (1.023)
Gender nonconformity	0.676 (0.316) *	0.655 (0.309) *	0.608 (0.310) *	0.624 (0.313) *
Perceived stigma from family members		0.480 (0.070) ***		
Sexual orientation microaggression			0.247 (0.039) ***	
Internalized sexual stigma				
Social discomfort				0.247 (0.108) *
Sexuality				0.139 (0.223)
Identity				0.476 (0.145) **

* *p* < 0.05; ** *p* < 0.01; *** *p* < 0.001.

**Table 4 ijerph-19-05780-t004:** Associations of problematic smartphone use with depression and anxiety: Stepwise multivariate linear regression analysis.

Variables	Depression	Anxiety
	B (SE)	B (SE)
Gender	−1.649 (0.744) *	−5.457 (0.942) ***
Sexual orientation	−1.464 (0.712) *	
Gender nonconformity	0.663 (0.222) **	
Sexual orientation microaggressions	0.241 (0.028) ***	0.230 (0.032) ***
Internalized sexual stigma		
Social discomfort	0.344 (0.059) ***	0.421 (0.078) ***
Sexuality		0.505 (0.173) **
Problematic smartphone use	0.160 (0.023) ***	0.153 (0.026) ***

* *p* < 0.05; ** *p* < 0.01; *** *p* < 0.001.

## Data Availability

The data will be available upon reasonable request to the corresponding authors.

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
