# Peer review of "Problematic Smartphone Use and Its Associations with Sexual Minority Stressors, Gender Nonconformity, and Mental Health Problems among Young Adult Lesbian, Gay, and Bisexual Individuals in Taiwan"

_ijerph, 2022, doi:10.3390/ijerph19095780_

Round 1

Reviewer 1 Report

This study of problematic smartphone use recruited an interesting sample of lesbian, gay and bi-sexual participants and covered a wide range of measures of stigma to look at the relationships with depression and anxiety. As such, the study is important and provides good information on variables allowing future studies to do comparisons. The authors duly note limits of the study and provide a good discussion. I have a few suggestions, the most important of which is the that the authors need to define problematic smartphone use from both a conceptual and measurement perspective.

For Model IV, the 3 dimensions of internalized sexual stigma should have high multiple collinearity as that would indicate good construct validity. Given that collinearity was not detected in the model, this raises suspicions. The authors should provide a correlation matrix of the model variables as this would improve the ability of future researchers to evaluate and better understand these complex relationships.

Please note in the discussion that LGB individuals who face SOMs may also avoid seeking treatment and/or they may assume that therapists would also have biases and engage in micro-aggressions.

In the conclusions, the authors could also mention the need for family and friend interventions. The pervasiveness of the stigma also indicates that interventions should address stigma on a cultural level and increase awareness of micro-aggressions and the harmful effects of such actions.

Author Response

We appreciated your valuable comments. As discussed below, we have revised our manuscript with underlines based on your suggestions. Please let us know if we need to provide anything else regarding this revision.

Comment 1

This study of problematic smartphone use recruited an interesting sample of lesbian, gay and bi-sexual participants and covered a wide range of measures of stigma to look at the relationships with depression and anxiety. As such, the study is important and provides good information on variables allowing future studies to do comparisons. The authors duly note limits of the study and provide a good discussion.

Response

Thank you for your positive comment.

Comment 2

I have a few suggestions, the most important of which is that the authors need to define problematic smartphone use from both a conceptual and measurement perspective.

Response

Thank you for your comment. In Introduction section of the revised manuscript, we added the contents defining and introducing problematic smartphone use from both a conceptual and measurement perspective. Please refer to line 37-59.

Griffiths firstly proposed the concept of technological addictions and operationally defined it as a behavioral addiction that involves human-machine interaction and is non-chemical in nature [1]. The Diagnostic and Statistical Manual of Mental Disorders, 5th edition (DSM-5) proposed the diagnostic criteria of Internet gaming disorder to define addiction to Internet gaming under the conditions for further study of Section III [2]. Smartphone addiction or problematic smartphone use (PSU) is another Smartphones are technological addiction warranted further examination [3-7]. Smartphones are an epoch-making invention. Smartphones can assist people in their daily lives by speeding up online information transfer, services, and interpersonal interaction as well as providing entertainment such as playing games, watching videos, and taking pictures. However, studies have found that PSU is consistently related to mental health problems (e.g., depression, anxiety, chronic stress, and low self-esteem) [8], physical discomfort (e.g., back, neck, and wrist pain) [9], and distracted driving in young adults [10]. Research has suggested that smartphone addiction or PSU should be conceptualized as a multi-dimensional construct. For example, Lin et al. proposed that smartphone addiction contains the dimensions of compulsive behavior, functional impairment, withdrawal and tolerance [3]. Kwon et al. proposed that smartphone addiction contains the dimensions of daily-life disturbance, positive anticipation, withdrawal, cyberspace-oriented relationship, overuse, and tolerance [11]. Both Lin et al. [3] and Kwon et al. [11] identified tolerance, withdrawal and function impairments as the essential characteristics of PSU, indicating that PSU shares the core symptoms with substance use disorders and Internet gaming disorder in DSM-5 [2] and called for further study on PSU in people of various groups.

Comment 3

For Model IV, the 3 dimensions of internalized sexual stigma should have high multiple collinearity as that would indicate good construct validity. Given that collinearity was not detected in the model, this raises suspicions. The authors should provide a correlation matrix of the model variables as this would improve the ability of future researchers to evaluate and better understand these complex relationships.

Response

Thank you for your comment. We added a correlation matrix of the model variables (Table 2) and described the correlation between problematic smartphone use and other factors as below. Moreover, we added condition index of all four regression models to present the level of collinearity as below.

Table 2 provides the bivariate correlations between the studied variables. Problematic smartphone use was significantly positively correlated with the following variables: perceived stigma from family members (r = 0.219; p < 0.001), sexual orientation microaggression (r = 0.202; p < 0.001), and internalized sexual stigma dimensions of social discomfort (r = 0.195; p < 0.001), sexuality (r = 0.139; p < 0.001) and identity (r = 0.215; p < 0.001).Please refer to line 233-238.

“Condition indices of Model I, Model II and Model II were 24.609, 27.731 and 27.745, indicating no problem of collinearity (<30). However, condition index of Model IV was 31.054, indicating that the problem of collinearity needed to be noticed.” Please refer to line 244-246.

Comment 4

Please note in the discussion that LGB individuals who face SOMs may also avoid seeking treatment and/or they may assume that therapists would also have biases and engage in micro-aggressions.

Response

Thank you for your suggestion. We added it as below into Discussion section. Please refer to line 287-290.

“LGB individuals who face SOMs may also avoid seeking treatment for PSU or other psychiatric disorders because of worrying about encountering SOMs from therapists. Delay treatment for PSU or other psychiatric disorders may further exacerbate PSU.”

Comment 5

In the conclusions, the authors could also mention the need for family and friend interventions. The pervasiveness of the stigma also indicates that interventions should address stigma on a cultural level and increase awareness of micro-aggressions and the harmful effects of such actions.

Response

Thank you for your suggestion. We added more contents as below into Conclusion section. Please refer to line 366-377.

Given that total acceptance is far from complete due to the continued prevalence of generally unwarranted negative familial and cultural stereotypes, further efforts to modify outdated beliefs with regard to LGB individuals are warranted. Constitutions, laws and anti-discrimination policies at the national level that include protection from discrimination on the grounds of sexual orientation is of fundamental importance [69]. Broadening understanding of LGB culture and awareness of prejudices toward LGB individuals in educational settings, workplaces, and family environments are also the necessary steps to help reduce sexuality-related stigma [69,70]. Public health strategies addressing attitudes to sexual orientation and promoting the changes of attitudes toward sexual minority among the general population may contribute to diverse affirmative cultural scripts regarding LGB individuals’ lives [69,71].

Reviewer 2 Report

Main message of the article
In the present study, the authors focused on problematic smartphone use (PSU) and associations of sexual minority stressors, gender nonconformity, depression, and anxiety in young adult lesbian, gay, and bisexual (LGB) individuals. Based on responses from 1,000 LGB individuals, results highlighted that perceived sexual stigma from family members, SOMs, internalized sexual stigma in the dimensions of social discomfort and identity, and gender nonconformity was significantly associated with PSU severity which was, in turn, significantly associated with depression and anxiety.

General Judgment Comments
The paper is written in a clear language and a fluent style. The design is appropriate; the abstract and the title are explanatory for the study. Results are reported following the required standards. The statistical analyses are appropriate and adequately explained. The Tables are informative and adding figures describing the models investigated would facilitate the comprehension of the analyses and results. The discussion section is very interesting and well-developed.

A few issues need to be addressed, as listed below.

Major Issues       

- I think that “problematic smartphone use” is a bit confounding since it appears that the usage derives from the need for social interactions (seeking a sexual partner, support); the authors should clarify better (i.e., in the Introduction and the discussion their choice to focus on PSU instead of problematic usage of social media.

Minor Issues       
- It would facilitate the reading to divide the Introduction into sections corresponding to the factors investigated in the study (PSU and LGB individuals, PSU and depression/anxiety or other psychological stressors, etc.).
- Could you please add information and distinguish how many men/women were homosexuals and how many were bisexual?
- Please add a table reporting the Cronbach’s alpha values for all the scales (including subscales) adopted for the study.
- Please add figures corresponding to Model I, Model II, Model III, and Model IV.
- Please divide the Discussion section into subparagraphs corresponding to the variables or models investigated in the study.
- Please add more information concerning the practical implication of the study.

Final comments
The author should edit the article following the comments listed above and resubmit the manuscript for further consideration.

Author Response

We appreciated your valuable comments. As discussed below, we have revised our manuscript with underlines based on your suggestions. Please let us know if we need to provide anything else regarding this revision.

Comment 1

General Judgment Comments
The paper is written in a clear language and a fluent style. The design is appropriate; the abstract and the title are explanatory for the study. Results are reported following the required standards. The statistical analyses are appropriate and adequately explained. The discussion section is very interesting and well-developed.

Response

Thank you for your positive comment.

Comment 2

The Tables are informative and adding figures describing the models investigated would facilitate the comprehension of the analyses and results.

Response

Thank you for your comment. We added a figure illustrating four multivariate linear regression models that examined the factors related to problematic smartphone use. Please refer to line 219.

Comment 3

Major Issues       

- I think that “problematic smartphone use” is a bit confounding since it appears that the usage derives from the need for social interactions (seeking a sexual partner, support); the authors should clarify better (i.e., in the Introduction and the discussion their choice to focus on PSU instead of problematic usage of social media.

Response

  1. Thank you for your comment. In Introduction section of the revised manuscript, we added the contents defining and introducing problematic smartphone use from both a conceptual and measurement perspective and explained the multiple purposes of smartphone use.
  2. We did not survey the purposes of smartphone use. Various purposes of smartphone use may contribute various levels of reinforcements and severities of PSU. We listed it as one of the limitations of this study as below.

Griffiths firstly proposed the concept of technological addictions and operationally defined it as a behavioral addiction that involves human-machine interaction and is non-chemical in nature [1]. The Diagnostic and Statistical Manual of Mental Disorders, 5th edition (DSM-5) proposed the diagnostic criteria of Internet gaming disorder to define addiction to Internet gaming under the conditions for further study of Section III [2]. Smartphone addiction or problematic smartphone use (PSU) is another Smartphones are technological addiction warranted further examination [3-7]. Smartphones are an epoch-making invention. Smartphones can assist people in their daily lives by speeding up online information transfer, services, and interpersonal interaction as well as providing entertainment such as playing games, watching videos, and taking pictures. However, studies have found that PSU is consistently related to mental health problems (e.g., depression, anxiety, chronic stress, and low self-esteem) [8], physical discomfort (e.g., back, neck, and wrist pain) [9], and distracted driving in young adults [10]. Research has suggested that smartphone addiction or PSU should be conceptualized as a multi-dimensional construct. For example, Lin et al. proposed that smartphone addiction contains the dimensions of compulsive behavior, functional impairment, withdrawal and tolerance [3]. Kwon et al. proposed that smartphone addiction contains the dimensions of daily-life disturbance, positive anticipation, withdrawal, cyberspace-oriented relationship, overuse, and tolerance [11]. Both Lin et al. [3] and Kwon et al. [11] identified tolerance, withdrawal and function impairments as the essential characteristics of PSU, indicating that PSU shares the core symptoms with substance use disorders and Internet gaming disorder in DSM-5 [2] and called for further study on PSU in people of various groups. Please refer to line 37-59.

Last, we did not survey the purposes of smartphone use. LGB individuals may use smartphones for various purposes, for example, exchanging messages, searching for partners, playing games, and watching videos. Various purposes of smartphone use may contribute various levels of reinforcements and severities of PSU. Please refer to line 355-358.

Comment 4

Minor Issues       
- It would facilitate the reading to divide the Introduction into sections corresponding to the factors investigated in the study (PSU and LGB individuals, PSU and depression/anxiety or other psychological stressors, etc.).
Response

Thank you for your comment. We divided the Introduction into sections by adding “1.1. Problematic Smartphone Use and Its Associations with Health Problems(line 36),1.2. PSU in Lesbian, Gay, and Bisexual Individuals(line 60), 1.3. PSU and Sexual Minority Stressors(line 76),1.4. PSU and Gender Nonconformity(line 98) and “1.5. Aims of This Study” (line 109).

Comment 5

- Could you please add information and distinguish how many men/women were homosexuals and how many were bisexual?
Response

Thank you for your comment. We added the numbers of bisexual women, lesbians, bisexual men and gays into Table 1. We also introduced in the text as below. Please refer to line 223-224.

“29.5% was bisexual women, 20.5% was lesbians, 13.5% was bisexual men, and 36.5% was gays;”

Comment 6

- Please add a table reporting the Cronbach’s alpha values for all the scales (including subscales) adopted for the study.
Response

Thank you for your comment. We added the Cronbach’s alpha values for all the scales into Table 1. Please refer to line 223-224.

Comment 7

- Please add figures corresponding to Model I, Model II, Model III, and Model IV.
Response

Thank you for your comment. We added a figure illustrating four multivariate linear regression models that examined the factors related to problematic smartphone use. Please refer to line 219.

Comment 8

- Please divide the Discussion section into subparagraphs corresponding to the variables or models investigated in the study.
Response

Thank you for your suggestion. We divided the Discussion section into subparagraphs titled “4.1. PSU, Perceived Sexual Stigma from Family Members, SOMs and Internalized Sexual Stigma(line 273), 4.2. PSU and Gender Nonconformity” (line 320), 4.3. PSU, Depression and Anxiety” (line 331), and “4.4. Limitations” (line 346).

Comment 9

- Please add more information concerning the practical implication of the study.

Response

We added the practical implication of the study as below into the revised manuscript. Please refer to line 366-377.

Given that total acceptance is far from complete due to the continued prevalence of generally unwarranted negative familial and cultural stereotypes, further efforts to modify outdated beliefs with regard to LGB individuals are warranted. Constitutions, laws and anti-discrimination policies at the national level that include protection from discrimination on the grounds of sexual orientation is of fundamental importance [69]. Broadening understanding of LGB culture and awareness of prejudices toward LGB individuals in educational settings, workplaces, and family environments are also the necessary steps to help reduce sexuality-related stigma [69,70]. Public health strategies addressing attitudes to sexual orientation and promoting the changes of attitudes toward sexual minority among the general population may contribute to diverse affirmative cultural scripts regarding LGB individuals’ lives [69,71].

Comment 10

Final comments
The author should edit the article following the comments listed above and resubmit the manuscript for further consideration.

Response

We revised the manuscript based on your suggestions and resubmitted it. We appreciate your valuable comments.

Round 2

Reviewer 2 Report

No further comments on my side. 
All the best to the authors for their future studies in the field.